# Gut Symptoms, Gut Dysbiosis and Gut-Derived Toxins in ALS

**DOI:** 10.3390/ijms25031871

**Published:** 2024-02-03

**Authors:** Aven Lee, Robert Henderson, James Aylward, Pamela McCombe

**Affiliations:** 1Centre for Clinical Research, The University of Queensland, Brisbane, QLD 4029, Australia; robert_henderson@health.qld.gov.au (R.H.); pamela.mccombe@uq.edu.au (P.M.); 2Department of Neurology, Royal Brisbane & Women’s Hospital, Brisbane, QLD 4029, Australia; 3Wesley Research Institute, The Wesley Hospital, Auchenflower, QLD 4066, Australia; jim.aylward@oncolin.com

**Keywords:** amyotrophic lateral sclerosis, gut microbiota, dysbiosis, excitotoxicity, formaldehyde, D-serine, homocysteine, homocysteinesulfinic acid

## Abstract

Many pathogenetic mechanisms have been proposed for amyotrophic lateral sclerosis (ALS). Recently, there have been emerging suggestions of a possible role for the gut microbiota. Gut microbiota have a range of functions and could influence ALS by several mechanisms. Here, we review the possible role of gut-derived neurotoxins/excitotoxins. We review the evidence of gut symptoms and gut dysbiosis in ALS. We then examine a possible role for gut-derived toxins by reviewing the evidence that these molecules are toxic to the central nervous system, evidence of their association with ALS, the existence of biochemical pathways by which these molecules could be produced by the gut microbiota and existence of mechanisms of transport from the gut to the blood and brain. We then present evidence that there are increased levels of these toxins in the blood of some ALS patients. We review the effects of therapies that attempt to alter the gut microbiota or ameliorate the biochemical effects of gut toxins. It is possible that gut dysbiosis contributes to elevated levels of toxins and that these could potentially contribute to ALS pathogenesis, but more work is required.

## 1. Introduction

Amyotrophic lateral sclerosis (ALS) is neurodegenerative disease, characterized by the loss of motor neurons [1]. Many possible pathogenetic mechanisms have been suggested in ALS. These include excitotoxicity and the accumulation of protein aggregates [2,3]. The development of ALS is thought to be multi-factorial, involving genetic and non-genetic factors, possibly through a multi-stage process [4,5,6,7]. Much is known about the genes associated with ALS [8]. Less is known about non-genetic factors, but epidemiological studies have identified smoking, dietary factors, heavy metals and pesticide exposure as possible risk factors for ALS [9]. 

Gut microbiota is now recognized as important in human health [10]. Dysregulation of the gut microbiota is another potential source of non-genetic factors that could influence the development of ALS [11,12]. Here, we review the current evidence for gut dysfunction and gut dysbiosis in ALS. We then examine a possible role of certain circulating gut-derived neurotoxins in ALS by reviewing how they are toxic to neurons, evidence for their association with ALS and the existence of pathways for their production by the gut microbiota; additionally, we review possible therapeutic approaches to gut dysbiosis.

## 2. Gut Symptoms and Gut Dysbiosis in ALS 

A wide range of gastrointestinal (GI) symptoms occur in ALS patients. A recent study [13] investigated the prevalence of GI symptoms in 43 ALS patients and assessed whether there is an association of GI symptoms with the severity of disease, as determined by the ALSFRS-R score. Constipation (60.5%), rectal tenesmus (57.5%), hard stools (55.0%) and borborygmus (42.5%) were the most frequent GI symptoms in the cohort, and moderate–advanced ALS was correlated with constipation, acid regurgitation, eructation, rectal tenesmus and dysphagia. Others have previously reported GI motor dysfunction in ALS, with marked delays in colonic transit times, delayed gastric emptying and anal sphincter abnormalities (reviewed in [14]). It now appears that patients with ALS experience GI symptoms in the earliest stages of disease, possibly even before diagnosis [15]. Some possible causes include altered diet, lack of exercise, inadequate fiber intake due to dysphagia and dehydration and the effects of anti-cholinergic medication [16,17]. Another possible cause of gut symptoms in ALS could be the direct involvement of the gastrointestinal tract in ALS pathology. In the TDP-43 mouse model of ALS, there are changes in the enteric nervous system with expression of TDP43 in the gut [18,19,20].

There are reports of gut dysbiosis in ALS [15,21,22,23]; however, the studies carried out to characterize gut microbiota alterations have given contrasting results, likely due to differences in study design, including small sample sizes, lack of age and sex matching of control groups and different experimental methodologies (reviewed in [24]). Our own study showed significant abnormalities in the gut microbiota in ALS and found that there was a possible effect on disease severity, with shorter survival in patients with a greater ratio of Firmicutes to Bacteroides species [25].

A large study [26] used a two-sample Mendelian randomization approach to examine the link between the gut microbiota and ALS in 20,806 patients and 59,804 controls. This method is used in epidemiological studies to assess whether there is a causal link between a risk factor and disease. This study found that OTU10032 unclassified Enterobacteriaceae species-level OTU and unclassified Acidaminococcaceae were associated with a higher risk of ALS. Furthermore, increased abundance of OTU4607_Sutterella and Lactobacillales order was found to be related to a higher susceptibility to ALS. These bacterial species were found to be related to levels of Gamma-glutamyl-related metabolites, suggesting that a glutaminergic, excitotoxic mechanism could be involved in the pathogenesis of ALS. These findings support a causative relationship between the gut microbiota and ALS. 

There have also been studies of gut dysbiosis in animal models of ALS. In one study, early changes in 11 distinct phylae of the gut microbiota were correlated with more aggressive progression in ALS mice; furthermore, supplementation with *Akkermansia muciniphila* ameliorated the symptoms and prolonged survival [27]. The beneficial effects of *Akkermansia muciniphila* supplementation was likely due to its ability to increase nicotinamide concentrations in the CNS of ALS mice. 

Another longitudinal study reported that alterations in gut microbiota in ALS mice were evident by 37 days of age, suggesting that dysbiosis preceded the onset of decline in muscle strength and coordination (64 days) and muscle atrophy (90 days) [28]. The shift in microbial composition was accompanied by changes in immune molecules in the spinal cord, including increased F4/80 and CD11 expression, suggesting that microglia were progressively activated as disease developed. 

A proxy for gut dysbiosis is antibiotic use, which is known to alter the balance of microbial species in the gut. A case–control study involving 2484 ALS patients and 12,420 age- and sex-matched controls showed that repeated antibiotic use was associated with increased risk of ALS [29]. Furthermore, ALS mice that have been repeatedly exposed to antibiotics develop more severe phenotypes and increased loss of motor neurons [27]. Together, these results provide evidence that GI symptoms and dysregulation of the gut microbiota are found in ALS.

Important evidence that supports the causative role the gut microbiota in ALS includes the recent study using C9orf72^−/− HARVARD^ mice, which observed a prevalence of Helicobacter spp., among others [30]. The investigators demonstrated the infiltration of neutrophils and other immune cells along with substantial microglial activation in the spinal cord of the C9orf72 mouse model. They also showed that reducing the microbiota burden with broad-spectrum antibiotics or fecal microbial transplants from a pro-survival environment attenuated the inflammatory phenotypes and significantly extended survival of the C9orf72 mice. The study highlights the therapeutic potential of microbiota regulation in the treatment of ALS. 

The cause of gut dysbiosis in ALS is not known and could occur before the onset of disease or secondary to disease. We speculate that, if gut dysbiosis is harmful in ALS, it is possible that dysbiosis before the onset of disease could contribute to disease pathogenesis and that dysbiosis after the onset of disease could exacerbate disease. We suggest that this needs further research. 

The composition of the gut microbiota is largely determined by dietary intake [31,32]. It is possible that, in some patients, gut dysbiosis due to diet could precede the development of ALS. In other patients, after the development of ALS, alterations in dietary intake could lead to dysbiosis. Alterations in dietary intake could occur in patients with bulbar dysfunction or in other patients with reduced appetite, which is associated with ALS [33]. 

It can also be speculated that neurodegeneration could directly lead to gut dysbiosis, possibly through an effect on gut motility (as outlined above). As already mentioned, in an animal model of ALS, gut dysbiosis occurs early in disease, presumably as a consequence of neurodegeneration [27]. Finally, it can be speculated that the immune changes in ALS, which occur in the brain and the peripheral immune system [34], could lead to changes in the gut microbiota, although this is not an area that has been researched. It is known that inflammation of the gut mucosa can be associated with gut dysbiosis [35].

There are many possible consequences of gut dysbiosis. These include increased toxin absorption from the environment due to increased intestinal permeability, increased production of bacterial toxins, immune system dysregulation and metabolic abnormalities, among others; these are discussed in detail in our previous review [36]. In the present review, we are focusing on the possible consequences of gut dysbiosis and particularly the possible production of toxins that could contribute to the progression of disease, as discussed below. 

## 3. Gut-Derived Toxins in ALS

The work described above supports an association of gut dysbiosis and mentions several mechanisms by which dysbiosis could contribute to disease. We now consider the possibility that gut-derived toxins contribute to ALS in some patients. Molecules such as formaldehyde (FA), D-serine and homocysteinesulfinic acid (HCSA) can be produced in the gut from dietary constituents, can be transported from the gut to the blood and from the blood to the brain and are toxic to neurons. We now review the evidence that these molecules are neurotoxic, review available epidemiological associations with ALS and review how these molecules can be produced in the gut and travel from the gut to the CNS. These issues are summarized in Table 1. We then present the evidence that these toxins are present in subjects with ALS (Table 2). This comes from our own work [37,38,39] and from the work of others [40,41,42]. 

### 3.1. Formaldehyde (FA) and ALS

#### 3.1.1. Neurotoxicity of FA

FA is a small molecule that is an environmental pollutant and is also generated during metabolism, through the one-carbon cycle [60]. FA can induce neurotoxicity by its ability to form cross-linking with DNA, proteins and lipid molecules [43]. FA can freely cross the blood–brain barrier (BBB) and is thought to have direct neurological effects [44]. In particular, FA increases mitochondrial membrane permeability [61] and causes oxidative damage via dysregulation of superoxide dismutase [62] as well as by inducing Tau protein misfolding and aggregation [63], which are mechanisms that are implicated in ALS [64]. 

#### 3.1.2. ALS Risk Linked to FA

Exposure to FA has been implicated in the risk of development of ALS. The Cancer Prevention Study II surveyed more than 1 million participants to obtain information about chemical exposures and found an increased rate of ALS among those who reported exposure to FA. When the analysis was restricted to individuals who reported their duration of FA exposure, a significant trend was found with increasing years of exposure [45]. A later study found that individuals with “high intensity” exposure to FA had almost four times higher risk of developing ALS than those with no exposure [46]. More recently, a detailed meta-analysis of human epidemiological studies of FA, followed by bioinformatic analysis using WikiPathways and the Comparative Toxicogenomics Database, identified pathways that may be highly affected by FA [47]. This analysis revealed that high exposure to FA increased ALS meta risk by 78% (meta-RR = 1.78); individuals with higher exposure to FA had a higher meta-RR in comparison to those with low FA exposure. Biological pathways found to be affected by FA exposure included folate and vitamin B12 metabolism pathways and the oxidative stress pathway. These findings suggest that exposure to FA could contribute to the initiation and progression of ALS by dysregulating common pathways. 

#### 3.1.3. Production of FA by Gut Microbes 

FA is known to be produced by some gut microbes [65]. Gut microbiota can metabolize dietary constituents to produce small molecule precursors to FA. Dietary quaternary amines (L-carnitine, betaine and choline) can be metabolized to trimethylamine (TMA), which is converted to trimethylamine N-oxide (TMAO) followed by subsequent conversion to dimethylamine (DMA) and FA. Alternatively, TMA can undergo N-demethylation to produce DMA and FA [65]. Microorganisms such as methylotrophs can further convert DMA to methylamine (MA) and then ammonia, each step producing FA. Since FA is a small molecule, it can be transported by diffusion. 

#### 3.1.4. Evidence of Increased FA in ALS Patients

We quantified the levels of FA in 40 healthy controls and 50 ALS patients using a FA detection assay (Table 2) [39]. The ALS patients in our study had no history of occupational exposure to FA. We performed targeted mass spectrometry to quantify levels of gut-derived metabolites TMA and TMAO, which are known precursors of FA. Plasma levels of FA were found to be significantly increased in ALS patients with ~30% of the patient cohort having 2–3-fold higher levels than those of control subjects. FA levels did not correlate with disease duration or ALSFRS-R score. FA levels were found to demonstrate a positive correlation with the intermediate molecules TMA and TMAO, possibly consistent with synthesis of FA from gut-derived precursor molecules. TMA and TMAO, which are produced by the gut microbiota from dietary quaternary amines, have been implicated in other neurological disorders and in cardiovascular disease (reviewed in [66]). 

### 3.2. D-Serine and ALS

#### 3.2.1. Neurotoxicity of D-Serine

Amino acids that are incorporated into proteins are the L-enantiomers. D-enantiomers are also present and can have a role in disease [67]. Of the D-amino acids known to exist in mammals, D-serine is considered to be the most biologically active. In contrast to L-serine, D-serine is not incorporated into peptides or proteins, thus constituting a free amino acid pool [68]. D-serine is a neurotoxic molecule. It is an endogenous co-agonist/activator at the D-serine/glycine binding site in the NR1 subunit of the N-methyl-D-aspartate (glutamate) receptor (NMDAR) [69,70]. Activation of the NMDAR requires the binding of both glutamate and its co-agonist. Whilst glycine also works as a co-agonist, the binding affinity of D-serine to the site is ~3-fold stronger than that of glycine, due to additional hydrogen bonding [71]. By virtue of its ability to increase the binding affinity of glutamate (the major excitatory neurotransmitter in the CNS), D-serine can produce excitotoxicity without any change in glutamate. The toxicity of D-serine could also arise from oxidative damage to cells, induced by products of their metabolism such as hydrogen peroxide [72]. 

#### 3.2.2. Abnormal D-Serine Metabolism in ALS

D-serine accumulates progressively in the spinal cord of ALS mice, and this was also reported in patients with familial ALS (FALS) and sporadic ALS (SALS) [41]. In the mouse model of ALS, D-serine accumulation is due to a combination of increased D-serine producing enzyme (serine racemase) and reduced D-serine degrading enzyme (D-amino acid oxidase or DAAO). Importantly, many dominant negative mutations in the *DAAO* gene (e.g., R38H, R199W, R199Q and Q201R) have been identified in patients with FALS that exhibit classical motor symptoms of ALS [48,49,50]. Human DAAO is highly expressed in the CNS and plays a critical role in regulating D-serine levels via oxidative deamination of D-serine to keto acids. Reduced activity of DAAO could possibly contribute to the increase in D-serine and exacerbate the hyperactivity of motor neurons via NMDARs [73,74]. The loss of enzymatic function of certain ALS-causing *DAAO* variants has been extensively studied. For example, the R199W/Q substitutions were shown to alter protein conformation and abolish enzymatic activity, resulting in abnormally increased D-serine levels [75]. Functional studies on the G183R *DAAO* variant have shown that this produced an inactive enzyme that forms protein aggregates and significantly increases cellular D-serine levels and the D/(D + L)-serine ratio [76]. Thus, the G183R mutation resembles the effects of R199W/Q found in FALS. Recent studies have reported the number of *DAAO* variants is extensive (>20), although their structure–function relationships are yet to be established [51]. By employing bioinformatic analysis and structural dynamic simulations, the authors predicted that substitutions in some of the variants would render them enzymatically inactive due to disrupted key interactions of the active site with the cofactor flavin adenine dinucleotide (FAD). However, such genetic variants are considered rare (<1% of FALS) and unlikely to be the cause of excitotoxicity in the majority of patients with ALS. The same can also be said for the well-characterized R199W/Q substitutions, which have an allele frequency of 0.000229 in ALS cases. It has been noted that, while some of these rare variants could be tolerated, they may be lethal when present with other *DAAO* variants, thus exerting a synergistic negative effect on the development of disease [51].

#### 3.2.3. Production of D-Serine by Gut Microbes 

While D-serine can be produced endogenously by serine racemase (SR) [77], it can also be produced by gut bacteria including Firmicutes and Bacteroidetes. Studies have shown that free D-amino acids, including D-serine, and several others are produced by intestinal microbiota and have identified bacterial groups (genus Eisenbergiella, Clostridium XVIII and Coprobacillus), belonging to the phylum Firmicutes as the relevant bacterial candidates [78]. The ratio of Firmicutes and Bacteroidetes phyla is frequently regarded as an important index for the health of the GI tract in humans [79]. Patients with ALS are reported to have a higher percentage of Firmicutes and a lower percentage of Bacteroidetes, when compared with healthy controls [80], with shorter survival in patients with a greater ratio of Firmicutes/Bacteroides species [25].

Much of the brain D-serine is produced by SR from L-serine. However, SR-knockout mice demonstrate 80–90% decrease in D-serine levels in the cortex, hippocampus and striatum, indicating that up to 20% of the brain D-serine is produced by processes that are not dependent on SR [81]. We speculate that D-serine produced by the gut microbiota is a possible source of D-serine in the nervous system and can be transported to the brain from the intestinal lumen. D-serine in the systemic circulation has access to the CNS via transport mechanisms such as 4F2hc/LAT1, which is expressed at the BBB and has a preferred transport of the D-enantiomer [52,53]. Thus, elevated levels of D-serine in ALS could possibly be derived from the gut and could enter the CNS to cause toxicity. Because D-serine is an essential co-agonist for NMDAR activation, a high brain D-serine is expected to exacerbate excitotoxicity in ALS. 

#### 3.2.4. Evidence of Elevated D-Serine in ALS

We have investigated levels of D-serine in 30 healthy controls and 30 ALS patients. We used a sensitive LC-MS method that is able to separate D-serine from L-serine enantiomers (Table 2) [38]. We found that ~43% of the patient cohort had plasma D-serine levels that were 2–4-fold higher than those of healthy controls. D-serine levels were found to be significantly higher in patients with bulbar onset compared to those with lower limb onset. In ALS patients, plasma D-serine levels were not correlated with disease duration or ALSFRS-R score. ALS patients with higher concentrations of circulating D-serine had shorter survival than patients who exhibited normal levels; however, this was not statistically significant, possibly due to the small numbers. In light of our mass spectrometry findings in this study and the histological findings that showed increased D-serine in ALS spinal cord [41], D-serine overproduction could be a feature among ALS patients and deserves further study. It is also known that increased levels of glutamate remain unchanged during ALS disease progression, whereas levels of D-serine can progressively increase during the disease course, suggesting a role in disease progression. Furthermore, the vulnerability of motor neurons to NMDA is augmented by D-serine and reduced by inhibition of SR [41]. Together, these findings suggest that D-serine could be a determinant of glutamate excitotoxicity in ALS. 

### 3.3. Excitatory Sulfur Amino Acids (SAAs) and ALS

#### 3.3.1. Neurotoxicity of SAAs

In the metabolism of methionine to taurine, excitatory sulfur amino acids (SAAs) are formed; these include the aspartate analogues cysteic acid (CA), cysteine sulfinic acid (CSA) and the glutamate analogues homocysteic acid (HCA) and homocysteinesulfinic acid (HCSA) [82,83]. Homocysteine (Hcy), the precursor of HCSA and HCA, is also an SAA and an intermediate in the conversion of methionine to cysteine. 

The neuroexcitatory actions of CA, CSA, HCA and HCSA are similar to those of glutamate [84], which is implicated in the development of ALS. SAAs are widely considered to be *bona fide* neurotransmitter candidates. Studies in cerebral cortex slices demonstrate that each of the SAAs can activate metabotropic glutamate receptors (mGluRs) and whilst they have lower efficacy than glutamate, they were more potent agonist (with EC_50_ ~401–408 µM compared with 791 µM for glutamate). 

HCSA is the most potent agonist at mGluR1, mGluR2, mGluR4, mGluR5, mGluR6 and mGluR8 receptors [85]. Electrophysiological studies in hippocampal neurons demonstrate HCSA can also evoke activation of NMDAR [86]. These findings are consistent with SAAs having a physiological role as endogenous activators of metabotropic and ionotropic excitatory amino acid receptors. The finding that SAAs could exert a cytotoxic action in cerebral cortical neurons and that the toxicity could be blocked by glutamate receptor antagonists show that SAAs can act directly as excitotoxins subsequent to activation of glutamate receptors [87]. If the levels of SAAs are increased sufficiently under pathological conditions, then the hyperactivation of a variety of glutamate receptors by SAAs could cause neuronal death through excitotoxicity. 

Hcy exerts its adverse effects via direct and indirect mechanisms. Excess Hcy can undergo auto-oxidation to generate hydrogen peroxide, superoxide and hydroxyl radicals to induce oxidative stress [88,89]. Elevated Hcy (homocysteinemia) can stimulate cytosolic calcium accumulation [90,91,92,93], mitochondrial dysfunction [94] and apoptotic pathway activation [91], which are pivotal factors in neurodegeneration. Increased levels of Hcy could also potentiate glutamate receptor activity via its oxidative metabolites HCSA and HCA, which are mixed excitatory agonists of NMDA and non-NMDA receptors [87,95,96,97]. 

#### 3.3.2. Abnormal Metabolism of SAAs in ALS 

Diverse changes in free amino acids have been reported in the motor cortex of ALS patients. The most notable of these is the increase in levels of taurine in gray matter of the ALS precentral gyrus and in the white matter of the same area, being >2-fold higher than controls [98]. Taurine is the final product of the metabolic pathway of SAAs, suggesting that SAAs (the intermediates of this pathway) could also be increased in ALS. 

Hyperhomocysteinemia is hypothesized to contribute to the development of several neurodegenerative disorders, including ALS [99], AD [100] and PD [101]. Hyperhomocysteinemia is classified as either moderate (15–30 µmol/L), intermediate (30–100 µmol/L) or severe (>100 µmol/L) [102]. Several studies have reported high blood plasma Hcy levels in patients with ALS, with levels ranging from 11 to 46 µmol/L [54,55]. MS-targeted metabolomics have also identified significantly higher levels of Hcy in the plasma of patients with ALS compared to healthy controls [56,57]. The most common genetic risk factor for hyperhomocysteinemia is a C677T polymorphism in the methylenetetrahydrofolate (*MTHFR*) gene that encodes the enzyme necessary for metabolizing Hcy. The TT genotype of *MTHFR* was found to be associated with increased risk of sporadic ALS in German and Swiss population [103]. Another study reported that the C677T polymorphism is a genetic risk for ALS in women [104]. However, a study performed in an Italian population failed to show an association between the C677T polymorphism and the risk of ALS [105]. A recent meta-analysis suggested that the TT polymorphism could be a genetic risk for sporadic ALS in the Caucasian population [106]. If the association between hyperhomocysteinemia and ALS is proven to be causal, then reducing the levels of Hcy could potentially reduce the incidence of ALS. Apart from *MTHFR* variants, there are other possible mechanisms by which hyperhomocysteinemia could occur in patients with ALS, including impairments in one-carbon metabolism, environmental factors such as diet, deficiency in cofactors and production by the gut microbiota [99,107].

#### 3.3.3. Production of SAAs by Gut Microbes

SAAs are known to be synthesized and released from CNS tissues [83]. SAAs can also be produced in the gut by several phyla that comprise the majority of the intestinal microbiota [108]. Using immobilized metal affinity chromatography, cysteine dioxygenase (CDO) was previously purified to homogeneity from Bacillus subtilis, Bacillus cereus and Streptomyces coelicolor [108]. In these organisms, CDO plays a critical role in regulating the steady-state levels of cysteine and is important for providing oxidized metabolites of cysteine, including taurine and sulfate. Kinetic analysis of the bacterial CDO homologues show that they were indeed *bona fide* CDOs with enzymatic properties (pH optimum, Km, Kcat/Km and specific activities) very similar to those of mammalian CDO. Phylogenetic analysis of the bacterial CDO homologues show that they are well distributed among Actinobacteria, Proteobacteria and Firmicutes. These findings suggest that many Eubacteria are able to synthesize cysteine sulfinic acid (CSA) from amino acid precursors such as cysteine. The expression of CDO in the gut microbiota also raises questions about production of other SAAs, including HCSA which is a byproduct of Hcy oxidation. Thus, bacteria belonging to A. muciniphila, Subdoligranulum sp, Eubacterium sp, and Clostridiales family XIII were recently identified as the main producers of Hcy and are increased in the gut microbiota in Parkinson’s disease [107]. Levels of Hcy were also found to be significantly correlated with abundances of these bacteria species.

#### 3.3.4. Evidence of Elevated SAAs in ALS

We measured the concentrations of SAAs in plasma samples from 30 healthy controls and 38 ALS patients by targeted mass spectrometry (Table 2) [37]. Of the four SAAs that were investigated, only HCSA showed significantly higher levels in the ALS patients compared to controls. We found that ~50% of the patients had HCSA levels that were 1.5–2-fold higher than those of controls. Of note, we found significant differences in HCSA levels between patients with bulbar and spinal onset ALS. Plasma levels of HCSA in ALS patients were not correlated with age, ALSFRS-R score or duration of disease. 

These findings are important because HCSA is just as potent as glutamate as an agonist for the NMDA receptor. HCSA is cytotoxic when tested on cerebellar cortical neurons [87] and is active at all NR2 subunits in the low micromolar range; its potency is similar to that of glutamate, being most potent at NR1/NR2D NMDA receptors with an EC_50_ ~3 µM [109]. Among the four SAAs, HCSA is structurally the most similar to glutamate. It is conceivable, therefore, that elevations in HCSA could predispose individuals to excitotoxicity via alterations in glutamate receptor functioning. Future studies will be needed to determine whether increased HCSA levels are also seen in ALS patients with hyperhomocysteinemia and whether, via glutamate receptors, they influence other risk factors to contribute to the pathogenesis of the disease. 

As already mentioned, ALS patients have higher Hcy levels compared to age- and sex-matched controls [55]. Higher levels of Hcy were also shown to be correlated with ALS disease progression [99]. From our data, one might infer that increased Hcy is metabolized to HCSA by the gut microbiota. HCSA produced by gut microbes can enter the circulation via efficient transport systems in the GI tract. In fact, HCSA, CA, CSA and HCA are all substrates for the EAAT family of glutamate transporters [58] and EAAT3/EAAC1 is known to be expressed in the intestine [59]. HCSA in the circulation could enter the CNS and activate synaptic receptors including NMDAR and mGluR and cause excitotoxicity in some patients with ALS. 

### 3.4. β-N-methylamino-L-alanine (BMAA)-Like Molecules

#### 3.4.1. Neurotoxicity of BMAA 

BMAA is a small molecule that has the structure of an amino acid [110,111]; it is highly reactive and can exist in free and protein-bound forms [112]. It is the best-known toxin in ALS [113]. BMAA is neurotoxic and increased levels of BMAA in ALS patients could contribute to the pathogenesis of ALS through a number of mechanisms. BMAA can integrate into nascent protein structures during mRNA translation, competing with L-serine (an amino acid that shares high structural similarity with BMAA) to induce protein misfolding and aggregation, resulting in the build-up of “junk proteins” that send cells into programmed cell death [114].

In the presence of bicarbonate ions, BMAA forms BMAA-carbonate, which resembles L-glutamate and can act as a potent glutamate receptor agonist to induce excitotoxicity [115,116]. BMAA can inhibit the cystine/glutamate antiporter-mediated cystine uptake (XCT), which leads to depletion of glutathione and increased oxidative stress [117]. BMAA can induce mitochondrial dysfunction. Exposure of motor neuron cell lines to BMAA elicited significant decrease in oxidative phosphorylation, perturbed calcium homeostasis and exacerbated production of ROS [118]. BMAA that enters the circulation can be transported into the brain where it is metabolized by CYP to the neurotoxin formaldehyde [119].

#### 3.4.2. BMAA and ALS

Dietary exposure to BMAA is proposed to be a cause of ALS in Guam [110]. ALS prevalence rates and death rates for the Chamorro residents of Guam were up to 100 times that of developed countries [120,121]. This hypothesis is supported by detection of BMAA in patient brain tissues with ALS (but not in controls) from Guam [122]. Researchers have also found BMAA in autopsy brain tissues from some ALS cases in populations beyond Guam [123,124]. BMAA is naturally produced by all major groups of cyanobacteria (blue-green algae), and is believed to be widespread in fresh water and marine water bodies, giving the potential for exposure through consumption of contaminated foods, aerosolization of water and recreational activities. BMAA has been identified in the marine environment in America and Europe and is known to enter the food chain, leading to human consumption which could lead to possible neuronal toxicity [110]. Production of BMAA and BMAA-like molecules by gut microbes could potentially be another source of neurotoxicity in ALS. 

#### 3.4.3. Production of BMAA-Like Molecules by the Human Gut Microbiota

Cyanobacteria are present in the human gut micro-flora [125]. Usually, cyanobacteria comprise only a small component of the gut micro-flora. However, poor diet or disease could alter the balance of normal bacteria and pathogens, enabling overgrowth of cyanobacteria with production of neurotoxins such as BMAA [126]. The production of BMAA by representatives of the gut micro-flora has long been hypothesized as a possible pathway of chronic exposure to the neurotoxin [126,127]. One potential candidate for the production of cyanotoxins is the melainabacteria, a non-photosynthetic phylogenetic clade of divergent cyanobacteria that has been discovered in the human gut [128]. So far, over 10 melainabacteria genomes have been partially or wholly assembled from human samples, highlighting their genomic diversity [129].

In a recent longitudinal study of microbiota composition in ALS, the gut microbiota was studied in 50 ALS patients and 50 control subjects, matched for age, sex, geographical origin and dietary habits [130]. In this pilot phase I trial, the ALS patients received either a placebo or probiotic supplement to assess the impact of such on the gut microbiota and progression of disease. They found that members of the cyanobacteria phylum were significantly increased in the patient group compared to the control group. 

BMAA is also known to be coproduced with the neurotoxin 2,4-diaminobutyric acid (DAB), which is a structural isomer of BMAA [131,132]. Like BMAA, DAB also activates glutamate receptors, with the potential to cause excitotoxicity [133]. Apart from the well-characterized routes of exposure (e.g., via ingestion, dermal contact and bio-magnification), it has been proposed that DAB could be derived from the human gut microbiota [134]. A recent study characterized the profiles of gut microbial metabolites in subjects with colorectal adenomatosis, which is thought to be strongly driven by gut dysbiosis; this study identified DAB as a significant metabolite that differed between adenoma cases and non-adenoma controls, suggesting the interplay between specific bacteria in the gut and the metabolites that they produce is important in disease development [135]. At present, it is unclear whether co-exposure to BMAA and other cyanotoxins potentiates the development of disease. It is possible that a synergistic effect could be produced by exposure to more than one cyanotoxin at a time.

Gut microbes have an extensive capacity to transform dietary compounds into neurotoxins and this is evident in the metabolism of L-carnitine (also choline and betaine) into trimethylamine, which is subsequently demethylated into formaldehyde [65]. This mechanism of biotransformation is particularly relevant to microbial species such as the archaea; they possess anaerobic methylation function and have the potential to generate BMAA-like molecules such as methylated forms of L-serine. Methylated amino acids, including O-methyl serine and O-methyl threonine, have the capacity to join to tRNA and be erroneously incorporated into proteins [136,137,138]. Such errors which result from the mischarging of tRNA by these alternative amino acids could potentially result in truncated or misfolded protein and subsequent cell damage. Interestingly, novel unpublished data from our group, derived from targeted mass spectrometry, revealed the existence of a methylated L-serine molecule (O-methyl L-serine or OMLS) in the blood of some patients with ALS. As a small hydrophilic amino acid, OMLS is structurally similar to BMAA—both molecules share a rigid backbone, with amino and carboxyl groups and a conjoining methyl group. In this regard, it is reasonable to hypothesize that BMAA-like molecules could originate from cleavage and methylation of amino acids, as previously proposed [139]. Such findings suggest that deleterious microbes (cyanobacteria, archaea and possibly others) in the human gut could produce BMAA-like molecules. 

Cyanotoxins such as BMAA are notoriously difficult to detect in complex matrices such as blood. The literature has discussed chromatography issues that could potentially impede detection, including the separation of BMAA from its isomers and interfering molecules as well as nuances with ionization efficiency [140]. BMAA can form dimers and adducts with a variety of metal ions, which makes mass spectrometry detection of this toxin in its native form rather difficult [141]. This may explain why some studies have failed to detect BMAA in plasma samples from ALS patients as well as the possibility that BMAA concentrations may be below the detection limit of the analytical instruments used [142,143]. We suggest development of a robust method that is selective and sensitive enough for detecting BMAA and related isomers in certain matrices (e.g., plasma and sera) is necessary for obtaining reliable data. There is little doubt that, if BMAA were present in sufficient concentrations, it could exert multiple modes of neurotoxic activity. 

## 4. Approaches to Alter the Gut Microbiota or to Ameliorate the Effects of Gut Toxins 

If gut-derived toxins play a role in ALS, it is necessary to consider whether there are any possible therapeutic interventions, either to prevent or treat ALS. This could be of particular importance to presymptomatic gene carriers, in whom an intervention to slow the development of disease would be helpful. We now review studies of interventions designed to improve the gut microbiota or to reduce the harmful effects of dysbiosis in ALS.

### 4.1. Diet

Diet is thought to be the main factor that determines the constituents of the gut microbiota, and a healthy diet is recommended to include increased intake of vegetables and dietary diversity [10]. This could be important advice for presymptomatic carriers of ALS genetic variants.

There have been studies of diet in ALS patients. The COSMOS study found that higher intakes of antioxidants and carotenes from vegetables are associated with higher functional (ALSFRS-R) and breathing (FVC) scores [144]. Another study found that a diet with foods high in animal-derived fat and protein early in disease could prolong survival [145]. In both studies, favorable diets were rich in variety and would be expected to enrich the diversity of gut microbiota [146]. However, in patients with ALS, there is a need to maintain caloric intake, which must be balanced against any advice regarding a “healthy diet”, so more studies are required.

### 4.2. Pre-, Pro-, Post- and Synbiotics

In many diseases, there have been attempts to alter the composition of the gut microbiota by treatment with prebiotics and probiotics (Table 3). Prebiotics are components of non-digested food that can provide a beneficial effect by modulating the gut microbiota. An early study to have reported the beneficial effects of prebiotics was conducted in ALS mice [147]. Here, the administration of galactooligosaccharides in SOD1-G93A mice delayed disease onset, extended survival, reduced motor neuron loss and muscle atrophy and suppressed the inflammatory responses in the CNS. Treatment with galactooligosaccharides was also shown to increase the concentrations of vitamin B12, folate and reduced Hcy levels.

Other prebiotics that have been investigated in ALS include the omega-3 polyunsaturated fatty acids (PUFA) such as eicosapentaenoic acid (EPA), docosahexaenoic acid (DHA) and alpha linolenic acid (ALA). A prior longitudinal study based on 1,002,082 participants in five prospective cohorts suggested that higher dietary intake of omega-3 PUFA could delay onset of ALS [148]. They found that intake of omega-3 PUFA was associated with a significant (~34%) reduced risk of ALS and that dietary intake of ALA was also associated with lower risk of ALS.

A new study conducted among 449 participants in the EMPOWER clinical trial examined whether plasma concentrations of ALA and other PUFAs were associated with disease progression, specifically survival time and functional (ALSFR-S) decline [149]. This study showed that higher plasma ALA levels were significantly associated with longer survival and slower functional decline in patients with ALS. After adjusting for sex, age, BMI, symptom duration, family history of ALS and ethnicity, it was found that individuals with the highest levels of plasma ALA had a ~50% lower risk of death during the study period compared to those with lower ALA levels. The findings suggest that specific PUFAs, especially ALA, have a favorable effect on disease progression in ALS. A limitation of these studies is the lack of information on the overall diet of the participants, including other supplements/nutrients and total calorie intake, which could all be associated with survival time.

Studies conducted on ALS mice have shown differing results. One study showed a positive association between consumption of PUFA and risk reduction [150]. Dietary intake of DHA significantly increased survival and delayed motor dysfunction in SOD1-G93A male mice, but not in females. In the same way, weight loss in this ALS model was also significantly influenced by DHA intake in male mice, but not in females. DHA supplementation also led to an increased anti-inflammatory fatty acid profile and reduced levels of circulating TNF-α in male mice, suggesting that dietary DHA is able to modulate sex-associated neuroinflammatory constraints. The data suggest that males benefit from yet-unknown mechanisms that are dependent on dietary DHA contents. On the contrary, another study conducted in SOD1-G93A mice showed that EPA intake did not influence how the disease progressed [151]. In fact, EPA treatment initiated at the presymptomatic stage decreased the lifespan of ALS mice. Animals that were treated with EPA demonstrated increased cellular damage and vacuolization in the spinal cord. In addition, microglia in the spinal cord of ALS mice treated with EPA showed increased levels of 4-hydroxy-2-hexena, which is a highly toxic aldehyde oxidation product of omega-3 PUFA. Taken together, although epidemiological studies suggest that omega-3 PUFA intake may be beneficial, care should be taken when adopting the use of such before its efficacy is proven by scientific studies and clinical investigations.

Probiotics are live organisms that promote a beneficial effect on the host microbiota [152]. A detailed study showed that probiotic supplementation of *Akkermansia mucini-phila*, which is a gut bacterium with an important role in mucin degradation, ameliorated the symptoms of ALS in SOD1-G93A mice [27]. The beneficial effect of the gut supplemented *Akkermansia muciniphila* was associated with increased nicotinamide levels in the CNS of SOD1-G93A mice and it was also shown that nicotinamide levels are downregulated in the sera and CSF of ALS patients.

Studies in humans have shown that administration of a probiotic containing five lactic acid bacteria (*Streptococcus thermophilus*, *Lactobacillus fermentum*, *L. delbrueckii* subsp. *delbrueckii*, *L. plantarum* and *L. salivarius*) modulated the bacterial diversity in patients with ALS but did not produce changes that would mimic healthy individuals. However, the probiotic treatment had no effects on disease progression, as monitored by ALSFRS-R score [130].

A recent study showed that treatment with *Lacticaseibacillus rhamnosus* HA-114 suppressed motor neuron degeneration in a *C. elegans* model of ALS [153]. Using a combination of genetic, behavioral and imaging analyses, the researchers showed that β-oxidation of fatty acids was impaired in the ALS worm model and restored with HA-114 supplementation, indicating that neuroprotection from this probiotic resides in its fatty acid content. The study also identified two genes linked to the neuroprotective actions of the probiotics: *acdh-1* and *acs-20*. Both genes are found in humans and are known to play critical roles in fatty acid metabolism and mitochondrial β-oxidation, pointing to lipid metabolism as perhaps being a key mechanism linked to neuroprotection in ALS. Further studies are needed to determine the full mechanistic scope of HA-114′s neuroprotective effect in animal ALS models.

There have also been attempts to ameliorate the effects of gut dysbiosis by use of postbiotics, which replicate the effects of beneficial bacteria. Postbiotics are bio-active compounds produced by food organisms in the fermentation process; these include short-chain fatty acids (SCFAs), secreted polysaccharides, extracellular polysaccharides, microbial fractions, teichoic acid, peptidoglycan-derived muropeptides and pili-type structures. The ease of delivery and the safety of postbiotics are generally better compared to probiotics [154]. The most important benefits of postbiotics, especially for SCFAs, are their anti-inflammatory and antioxidant properties [155].

One study showed that oral administration of butyrate (a four-carbon SCFA) restored microbial homeostasis and prolonged the life span of ALS mice [23]. In both ALS mice and intestinal epithelial cells cultured from humans, butyrate treatment significantly reduced aggregation of SOD1-mutated proteins. Subsequent longitudinal metabolomics studies revealed that butyrate administration was able to enhance healthy metabolite expression [156].

The beneficial effects of postbiotic phenylbutyrate combined with tauroursodeoxycholic acid (also considered to be a postbiotic) in ALS patients was investigated in a recent phase 2, randomized, placebo-controlled trial involving 137 participants over a 24-week period [157]. These postbiotics were shown to reduce neuronal death in a preclinical model of ALS [158] and in other neurodegenerative diseases [159,160,161,162]. The small trial found that phenylbutyrate–tauroursodeoxycholic acid treatment resulted in a modest reduction in functional (ALSFR-S) decline in ALS patients with a median overall survival of 4.8 months longer. Plasma levels of neuroinflammatory markers YKL-40 and CRP were significantly reduced in ALS patients receiving phenylbutyrate–tauroursodeoxycholic acid treatment and correlated with disease progression [163]. A phase 3 trial (NCT05021536) will now investigate the effects of phenylbutyrate–tauroursodeoxycholic acid on survival in a larger number of ALS participants and for a longer duration. These findings suggest that the use of metabolites derived from microorganisms (i.e., postbiotics) may be an attractive therapeutic strategy in ALS.

A potential area for future research is the personalized combination of prebiotics and probiotics (also called synbiotics) to regulate the microecology of the gut microbiota and bring benefits to the host [164]. No studies have investigated the effects of synbiotics on ALS animal models or ALS patients. However, there have been some studies of synbiotics in other neurodegenerative diseases that suggest they may have beneficial effects. Thus, synbiotics treatment of AD mice was shown to improve cognitive symptoms and markedly reduce levels of TNF-α [165]. In PD mice, synbiotics treatment reduced dopaminergic neuronal loss, improved blood–brain barrier integrity and increased BDNF and GDNF expression [166].

**Table 3 ijms-25-01871-t003:** Summary of pre-, post- and postbiotic studies conducted in ALS patients and models of ALS.

Subjects	Type of study	Agent	Effect	Ref
SOD1-G93A mice	Prebiotic trial	Galactooligosaccharides (GOS)	GOS treatment delayed onset of disease and extended survival of G93A mice; motor neuron loss and muscle atrophy were significantly reduced; increased blood concentrations of folate and vitamin B12 and reduced homocysteine level; suppressed microglia activation and regulated several inflammatory (iNOS and TNF-α)- and apoptosis (caspase-3, Bcl-2, and Bax)-related molecules.	[147]
ALS patients	Prebiotic association study	Omega-3 polyunsaturated fatty acids (ω-3 PUFA)	Higher ω-3 PUFA intake associated with a reduced risk of ALS, delayed onset.	[148]
ALS patients	Prebiotic association study	Alpha-linoleic acid (ALA)	Higher ALA intake found to be associated with longer survival and slower functional decline.	[149]
SOD1-G93A mice	Prebiotic trial	Docosahexaenoic acid (DHA)	DHA intake increased survival, delayed motor dysfunction and slowed weight loss in G93A male mice but not in females; increased anti-inflammatory fatty acid profile and reduced proinflammatory cytokine TNF-α in male mice.	[150]
SOD1-G93A mice	Prebiotic trial	Eicosapentaenoic acid (EPA)	EPA intake initiated at disease onset did not alter disease presentation and progression; EPA induced shorter lifespan. Microglia in spinal cord showed increased levels of 4-hydroxy-2-hexenal.	[151]
SOD1-G93A mice	Probiotic trial	*Akkermansia muciniphila* (AM)	AM intake substantially improved motor function and prolonged life span in G93A mice. AM attenuates ALS in G93A mice by increasing nicotinamide levels. AM treatment reduced brain atrophy.	[27]
ALS subjects	Probiotic trial	Five lactic acid bacteria: *Streptococcus thermophilus*, *Lactobacillus fermentum*, *L. delbrueckii* subsp. *delbrueckii*, *L. plantarum* and *L. salivarius*	Probiotic treatment did not bring intestinal microbiota of ALS patients closer to that of control subjects. No effects on disease progression.	[130]
*C.elegans* ALS model	Probiotic trial	*Lacticaseibacillus rhamnosus* HA-114	HA-114 treatment reduced motor neuron degeneration; slowed rate of progressive paralysis.	[153]
ALS mice	Postbiotic trial	Butyrate	Butyrate treatment prolonged survival, reduced serum levels of inflammatory cytokines (IL-7 and LPS) and suppressed microglia IBA1 level.	[156]
ALS patients	Postbiotic trial	Phenylbutyrate plus taurodeoxycholic acid (PB-TUDCA)	PB-TUDCA treatment reduced rate of decline in ALSFRS-R; reduced blood plasma concentrations of neuroinflammatory markers (YKL-40 and CRP) which correlated with disease progression.	[157,163]

A synbiotics approach could also be considered for ALS. For example, if cyanobacteria or cyanobacteria produced BMAA are important in ALS and are present in the human gut, then it would be helpful to identify microorganisms that are protective. One such candidate is the proteobacteria, which are known cyanobacterial toxin degraders. Of the identified strains capable of degrading microcystin or that carry the *mlrA* gene (a gene closely linked to known species/genera capable of degrading cyanotoxins), most belong to members of the α-proteobacteria [167]. These organisms also break down other gut cyanobacteria produced toxins such as anatoxin and saxitoxin that could also contribute to ALS. Proteobacteria are also potential butyrate producers since they express genes that encode enzymes that synthesize butyrate, including butyryl CoA dehydrogenase, butyryl CoA transferase and/or butyrate kinase [168]. Therefore, if α-proteobacteria are important protective bacteria in the human gut then such microbes can be targeted in future (synbiotics) therapy in order to modify the gut microbiota to alleviate disease progression and improve outcomes. In this context, it may be relevant to consider Lactobacillus supplements and prebiotics if α-proteobacteria are important. Lactobacillus are safe [169] and considered the best probiotic for restoring compositions of the gut microbiota and provide beneficial functions to the host through immune and neuro modulation [170,171]. In AD mice, *L. plantarum* MTCC1325 supplementation significantly reduced pathological hallmarks such as amyloid plaques and tangles as well as ameliorating cognition deficits [172]. Other similar research has shown that *L. acidophilus*, *B. bifidum* and *B. longum* restored synaptic plasticity and improved the impaired spatial cognitive performance in AD mice [173]. Studies have also supported the role of *L. acidophilus* LA02 and *L. salivarius* LS01 in reducing levels of reactive oxygen species and inflammatory cytokines (TNF-α, IL-6 and IL-17A) and increasing ant-inflammatory cytokines (IL-4 and IL-10) in peripheral blood mononuclear cells, thereby relieving PD-associated symptoms [174].

### 4.3. Therapies to Alter Metabolic Pathways

Another approach to ameliorate the effects of toxins produced by gut dysbiosis is to use therapies that alter the biochemical pathways involved in neurotoxicity. Potential therapeutic options exist that reduce HCSA, e.g., supplementation with folate and vitamin B12, have been shown to reduce Hcy levels in Alzheimer’s disease and Parkinson’s disease (reviewed in [175]), and this could also slow down the oxidation of Hcy to HCSA in ALS. Folate and vitamin B12 convert Hcy to methionine, thereby reducing levels of Hcy. A recent phase 3 randomized clinical trial showed that methylcobalamine B12 in high dose (50 mg) was efficacious in slowing functional decline in patients with early-stage ALS with moderate progression rate and was safe to use [176]. The trial showed that levels of Hcy in plasma were significantly reduced following methylcobalamine B12 supplementation. These encouraging results suggest that high-dose methylcobalamine B12 could be a promising agent to use.

There are other potential strategies such as treatment with anti-HCSA antibodies. For example, homocysteic acid (HCA) is considered to be pathological biomarker in the AD brain and increased levels of such are thought to induce neuronal amyloid beta peptide formation [177]. AD mice fed with vitamin B6-deficient food have high levels of HCA which accelerates pathological changes. In these animals, HCA toxicity was markedly reduced following vaccine treatment with anti-HCA antibody, which induced strong neurogenesis in the brain and recovery of memory performance [178]. These findings support the idea that oxidized metabolites of Hcy could potentially be etiological agents and an accelerator in the pathogenesis of neurodegenerative disease. It also highlights the potential benefits of an antibody-based approach in the treatment of neurodegeneration caused by HCA or HCSA toxicity.

## 5. Discussion

We have reviewed current information about possible gut dysbiosis and gut-derived toxins in ALS. First, we note that gut symptoms are present in subjects with ALS and that there is evidence of gut dysbiosis in ALS. The cause of gut dysbiosis in ALS is unclear. Diet is the main driver of gut dysbiosis and it is possible that dysbiosis precedes the onset of ALS. It is also possible that gut dysbiosis is a consequence of ALS, either due to changes in diet or possibly secondary to neurodegenerative processes, or the response to neurodegeneration. There are many possible consequences of gut dysbiosis and these have previously been reviewed [36]. These include changes in gut permeability, increased passage of dietary toxins, alterations in metabolism, alterations of immune responses due to interactions of the microbiota with the gut associated lymphoid tissues, effects on the nervous system and production of neurotransmitters by the gut as well as alterations in the gut–brain axis. It is possible that gut dysbiosis contributes to or exacerbates the pathogenesis.

We then focus on a possible role of gut-derived toxins. We review published information about the toxicity of FA, D-serine and SAAs. For FA, there is strong evidence that it is neurotoxic, that occupational exposure is associated with ALS, and there are mechanisms for production of FA by gut microbiota. We have studied FA levels in subjects with ALS, and found significant elevation in some patients. Importantly, we found that levels of gut microbial metabolites TMA and TMAO were greater in ALS subjects with elevated FA than those with normal levels. Another prior study also reported that the TMAO metabolic pathway of the gut microbiota is perturbed in patients with ALS [179]. These findings lead to the notion that imbalance of intestinal FA metabolism by gut microbiota may have a link to ALS.

For D-serine, we have reviewed the evidence that D-serine is neurotoxic, is increased in the brain and spinal cord of people with ALS and in animal models of ALS and could be produced by the gut microbiota. We also show that there are elevated levels of D-serine in some patients with ALS. While D-serine is produced endogenously in humans, it is also absorbed from gut bacteria along with D-glutamate [180,181,182]. Another example of a potential role of D-amino acids in neuropathology is the D-isomer of BMAA, which is derived from cyanobacteria. Studies in mice show that administration of the L-isomer of BMAA resulted in significant production of D-BMAA in the liver, circulation and the brain, suggesting that a mechanism for microbial interconversion between BMAA isomers might exist [183]. Toxicity investigations indeed showed that D-BMAA caused neuronal cell death in a dose-dependent manner, which was mediated through AMPA glutamate receptors. Hence, there is potential for a role of D-amino acids produced by the gut microbiota in the initiation and progression of ALS. D-amino acids were long considered irrelevant in mammalian biology, but accumulating data now show that they actually play an important part of human (patho)physiology. The gut microbiota has a large genetic capacity to produce D-amino acids [78]. The L– to D- conversion is catalyzed by specific racemases which are expressed by many bacterial species residing in the intestines of mammalian hosts [184]. Accordingly, the lumen of the GI tract is likely to be rich in free D-amino acids that can be derived from such bacteria. The presence of a number of microbial racemases raises a further question concerning what other non-protein amino acids may be converted from their L– to D-isomers that may potentially cause adverse effects. As emerging data have identified a relationship between gut microbiota dysbiosis and ALS, it is likely that D-amino acid dependent processes will become exceeding dysregulated.

The SAAs are a family of related molecules. We have reviewed the evidence that these molecules or their metabolites are neurotoxic, citing work that reports the accumulation of taurine (which is an SAA) in the ALS brain and work demonstrating that certain bacterial phyla in the human gut do indeed express *bona fide* enzymes for the biosynthesis of SAA. We have reviewed our own study that showed increased concentrations of HCSA in the blood of patients with ALS and studies that showed high levels of Hcy in ALS. These findings confirm metabolic abnormalities concerning SAA exist in ALS patients, suggesting that these excitotoxins may play a role in neurodegeneration in ALS. SAA metabolism is crucial for regulating the availability of methionine (a source of the methyl groups needed for all methylation reactions) and for protein homeostasis [185]. Consequently, impaired SAA metabolism leading to increased Hcy and HCSA in the blood plasma is likely to lead to overproduction of reactive oxygen species, increased oxidative stress, mitochondrial dysfunction and excitotoxicity.

Taken together, these studies show elevated levels of neurotoxins that have already been linked to ALS and could be produced by the gut microbiota. It is possible that these molecules are produced by the gut microbiota of ALS patients and contribute to pathogenesis (Figure 1). These molecules were only elevated in some patients. This is not unexpected, because ALS is heterogenous. Furthermore, there are likely to be multiple steps in the pathogenesis of ALS, leading to eventual death of motor neurons, and these steps are not necessarily the same in all patients.

We reviewed therapies to reduce gut dysbiosis in ALS and therapies to correct biochemical pathways that could be disturbed by gut-derived toxins. There are few such studies but there have been some promising studies with pre-, pro- and postbiotics and with high-dose vitamin B12 supplementation.

We acknowledge limitations in the interpretation of these data that report increased levels of gut toxins. The sample size for ALS patients was small. There is no information regarding the diets of the ALS patients and control subjects; this would be important if these molecules were produced by the gut microbiota, since diet can alter the composition of gut microbes. Dietary changes are known to occur during disease evolution, especially for bulbar-onset ALS, where patients often reduce the quantity and change the quality of food and liquid that are consumed [186]. All these changes could impact the gut microbiota [187]. Importantly, there was no measurement of the gut microbiota in the subjects with increased levels of neurotoxins, so that a direct association of levels of toxins with gut dysbiosis was not shown. Furthermore, these studies have not been replicated by others.

To confirm our suggestion that gut-derived neurotoxins contribute to the pathogenesis of ALS, further work is required. The next stage will be to determine whether subjects with increased levels of these molecules have abnormalities of the gut microbiota that could produce these molecules, and to adjust for dietary intake. Large numbers of subjects are required because of the complexity of gut microbiota and the challenges of statistical analysis [36]. To determine whether increased FA, D-serine or HCSA are indeed related to gut microbiota changes, it will be necessary to correlate their levels with levels of gut microbes such as TMA-producing bacterial species (e.g., Methylotrophs), Firmicutes (e.g., genera belonging to Eisenbergiella, Clostridium XVIII and Coprobacillus) and those belonging to the Eubacteria phylum, respectively.

## 6. Conclusions

We have reviewed studies that showed evidence of possible gut-derived toxins in the blood of subjects with ALS. We review the evidence that these are neurotoxic, review the epidemiological evidence linking exposure to these molecules to ALS and review the evidence that these molecules can be produced by the gut microbiota. It could be the case that these molecules are involved in the multi-stage process of neurodegeneration. It is also possible that the presence of these molecules is a factor that increases disease severity.

It must be noted that the production of toxic molecules is not the only pathway by which gut dysbiosis could influence ALS—as we have reviewed, there are other mechanisms [36]. These molecules have been implicated in other neurodegenerative diseases [101,107,188,189,190,191,192,193,194], so are not specific to ALS.

This field is important because little is known about the non-genetic factors that lead to ALS. Non-genetic factors are important because they can potentially be modified, in contrast to genetic factors. There are some preliminary studies of interventions to alter the gut microbiota in ALS, but more information is required.

## Figures and Tables

**Figure 1 ijms-25-01871-f001:**
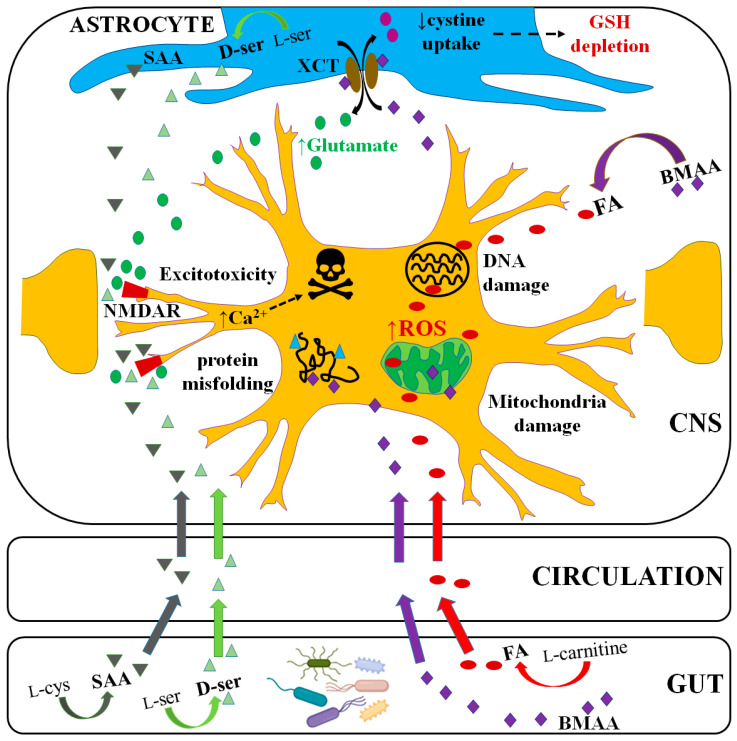
Schematic representation of endogenous neurotoxins (FA and BMAA) and excitotoxins (D-serine and HCSA) produced by the gut microbiota and transport into the brain to damage motor neurons. Toxic molecules produced by the members of the human gut microbiota can enter the circulation, be transported into the brain and have potential toxic effects on motor neurons through distinct mechanisms. Their actions on system XCT could play a central role, both causing oxidative stress (through depletion of GSH) and through overactivation of glutamate receptors to induce excitotoxicity. Other potential mechanisms are through misincorporation into proteins and mitochondrial damage via disruption of mitochondrial metabolism and ROS production. There is also the potential to cause DNA damage via formation of DNA adducts such as O6-mG that are mutagenic in cycling cells. Red, purple, light-green and dark-green symbols represent FA, BMAA-like molecules, D-serine and SAA, respectively, and colored arrows show their paths from the gut into the circulation and entry into the nervous system.

**Table 1 ijms-25-01871-t001:** Association of FA, D-serine and HCSA with ALS, their production by gut microbial species and mechanism of transport from gut to brain.

Molecule	Epidemiological Association with ALS	Toxicity	Microbial Production	Transport from Gut to Blood	Transport from Blood to CNS	References
FA	Increased rate of ALS among individuals with increased exposure to FA.	Neurotoxic	Methylotrophs, Firmicutes	Diffusion	Diffusion	[43,44,45,46,47]
D-serine	D-serine accumulates in spinal cord and blood of ALS patients who carry *DAO* gene mutations; allele frequency ~0.00029.	Excitotoxic	Firmicutes; genus: Eisenbergiella, Clostridium XVIII, Coprobacillus	ATB(0,+)	4F2hc/LAT1	[41,48,49,50,51,52,53]
HCSA	CSF levels of homocysteine (the precursor of HCSA) among ALS population are markedly higher than controls, indicating that homocysteine might play a potential role in the pathogenesis of disease.	Excitotoxic	Actinobacteria, Proteobacteria and Firmicutes	EAAT3	Transporter not yet identified	[54,55,56,57,58,59]

**Table 2 ijms-25-01871-t002:** Summary of studies investigating plasma levels of FA, D-serine and HCSA in ALS patients and healthy control subjects.

Molecule	Sample Size	Age (Mean ± SD)	ALSFRS-R	Disease Duration (Median, Range) (Days)	Detection Method	% of Patients with Increased Levels	Bulbar Onset vs. Spinal Onset	Correlation Clinical Features	Ref
FA	50 patients 40 controls	60.6 ± 7.7 57.5 ± 10.3	35.5 ± 6.11	633.5 (145, 2900)	FA fluorometric detection assay; LC-MS	~30% of patient cohort had 2–3-fold higher FA levels than controls	No significant difference	FA positively correlated with TMA and TMAO; no significant correlation of FA with age, ALSFRS-R or duration of disease	[39]
D-serine	30 patients 30 controls	60.3 ± 7.4 56.4 ± 10.4	35.7 ± 6.4	950 (839, 1660)	LC-MS	~43% of patient cohort had 2–4-fold higher D-serine levels than controls	Higher D-ser in bulbar onset patients	No significant correlation of D-serine with age, ALSFRS-R or duration of disease	[38]
HCSA	38 patients 30 controls	62.4 ± 8.5 58.1 ± 8.2	36.0 ± 5.4	889 (762, 1761)	LC-MS	~50% of patient cohort had 2–3-fold higher HCSA levels than controls	Higher HCSA in spinal onset patients	No significant correlation of HCSA with age, ALSFRS-R or duration of disease	[37]

## Data Availability

This is a review article. Data from the research papers that are cited are available on request.

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
