# Peer review of "Gut Symptoms, Gut Dysbiosis and Gut-Derived Toxins in ALS"

_ijms, 2024, doi:10.3390/ijms25031871_

Round 1

Reviewer 1 Report

Comments and Suggestions for Authors

In this paper, Lee, et al. write a Review of “Gut symptoms, gut dysbiosis and gut derived toxins in ALS”. This is a well-organized review of gut symptoms with ALS. It gives some clues for how does the spinal cord neurodegeneration that affect the gut function. I have some questions need to be revised. The details as follows:

Major concern,

The author should add more articles about how the neurodegeneration induces the gut microbial changed.

Does the neurodegeneration change the total microbial composition in the gut.

How does the neuron degeneration induce the immune responses that affect the gut microbial?

How does the microbial changed further induced the neurodegeneration.

Because most of ALS genetic mutation proteins expressed in all organs and cells, the gut symptom maybe due to the genes mutation in the gut caused the gut function changed that induced the gut microbial changed.

Minor concern,

Please added more literatures that related to ALS neurons degeneration that affected the gut microbial composition.

Comments on the Quality of English Language

The author should concise some paragraph. make it more easy to understand.

Author Response

Major concern,

The author should add more articles about how the neurodegeneration induces the gut microbial changed. Does the neurodegeneration change the total microbial composition in the gut.

Response: We thank the reviewer for this comment.

In the previous version of our paper, we reviewed the evidence for the existence of gut dysbiosis in ALS. In the section on gut dysbiosis we have now added additional information about the possible causes of gut dysbiosis in ALS, including whether these are primary or secondary.   

How does the neuron degeneration induce the immune responses that affect the gut microbial?

Response: We think this is a complex issue. We acknowledge that changes in the gut microbiota can affect the host immune system. The reviewer is asking whether changes in the immune system on ALS can affect the gut microbiota.  

We have addressed this in the discussion by noting that there are immune changes in ALS, that could be a response to neurodegeneration. These are changes in the brain and in the peripheral immune system. We have searched for papers that address the role of the host immune system on the gut microbiota and not found any information on this. We have added a paper that says that inflammation of the gut can lead to changes in the gut microbiota.  

How does the microbial changed further induced the neurodegeneration.

Response: We have enlarged the discussion to mention the possible ways by which gut dysbiosis could contribute to neurodegeneration. The possible deleterious effects of gut dysbiosis on ALS pathogenesis include changes in gut permeability, allowing the passage of dietary toxins, alterations in metabolism due to alterations in energy metabolism in the gut, alteration of immune responses due to interaction of the gut microbiota with the gut associated lymphoid tissue, effects on the enteric nervous system, production of neurotransmitter by the gut and alterations of the gut brain axis.  We have mentioned these and cited our extensive review.

In the present paper we focus on the possible role of gut-derived toxins and provide evidence that there are increased levels of such toxins in people with ALS.

Because most of ALS genetic mutation proteins expressed in all organs and cells, the gut symptom maybe due to the genes mutation in the gut caused the gut function changed that induced the gut microbial changed.

Response: We acknowledge this is the case and have added a citation about the changes in the gut in the tdp-43 animal model of ALS. We note that once established, gut dysbiosis could lead to the production of toxins.

Minor concern,

Please added more literatures that related to ALS neurons degeneration that affected the gut microbial composition.

Response: We have added a citation to gut changes in tdp43 model of ALS. We are not aware of any other papers about this. In the present review we are not focussed on the possible causes of gut dysbiosis, but rather on the specific possibility that gut dysbiosis could be associated with the presence of circulating toxins.

Commentson the Quality of English Language

The author should concise some paragraph. make it easier to understand.

Response: we have edited the paper extensively.

Reviewer 2 Report

Comments and Suggestions for Authors

In this work, authors make a review of potential roles of gut derived toxins on ALS pathogenesis. Results from microbiome are difficult to interpretate because the observed changes could be causative, a consequence of the underlying disease, or an epiphenomenon. However, authors provided interesting evidence of a plausible causal role of gut derived toxins in vitro and in vivo. Important evidence that would further support the causative role of microbiome is the study on pre-symptomatic carriers (C9orf72, for example): is there any reference on that in human? Notably, in doi: 10.1038/s41586-020-2288-7 the impact of microbiota diversity on disease pathogenesis is previous of disease onset.

Nevertheless, is important to be aware that dietary changes within disease evolution exist, specially for bulbar ALS, and might be a source of variability that authors should discuss. Authors propose that microbiota derived toxins trigger excitotoxicity and oxidative stress and might be a target to study. However, Riluzole and Edaravone (an anti-excitatory and an antioxidant, respectively) have modest effects on survival: why targeting these effects through microbiome will report better results than these stablished treatments?

Authors highlights the importance of prebiotics and postulate them as promising therapeutic molecules. In the case of n-3 PUFAs, the main effect is thought to be mediated by resolvins and not by the micrbiome (to my knowledge): why including them as prebiotics? Is there any evidence for that, such as reducing mentioned toxins? Moreover, authors should consider that in doi: 10.1371/journal.pone.0061626 EPA supplementation has opposite effects of DHA. This work should also be discussed because not all n-3 PUFAS are supposed to be beneficial, at least in mouse models.

There are some "-" in the middle of some words and also some characters in different font (some bacteria names).

Author Response

In this work, authors make a review of potential roles of gut derived toxins on ALS pathogenesis. Results from microbiome are difficult to interpretate because the observed changes could be causative, a consequence of the underlying disease, or an epiphenomenon.

Response: We have added discussion of the possible causes of gut dysbiosis. However, our main intention in this paper is to focus on the consequences of gut dysbiosis and the possibility that gut dysbiosis could lead to the production of toxins that could contribute to ALS pathogenesis.

However, authors provided interesting evidence of a plausible causal role of gut derived toxins in vitro and in vivo. Important evidence that would further support the causative role of microbiome is the study on pre-symptomatic carriers (C9orf72, for example): is there any reference on that in human? Notably, in doi: 10.1038/s41586-020-2288-7 the impact of microbiota diversity on disease pathogenesis is previous of disease onset.

Response: We thank the reviewer for this suggestion and have added this reference to section 2. There is no reference on that in humans.

Nevertheless, is important to be aware that dietary changes within disease evolution exist, especially for bulbar ALS, and might be a source of variability that authors should discuss.

Response: This has been added to the section on gut dysbiosis and to the discussion.

Authors propose that microbiota derived toxins trigger excitotoxicity and oxidative stress and might be a target to study. However, Riluzole and Edaravone (an anti-excitatory and an antioxidant, respectively) have modest effects on survival: why targeting these effects through microbiome will report better results than these stablished treatments?

Response: We acknowledge there are controversies and unknown mechanism linking the microbiota and ALS. We believe that microbial therapies may be beneficial for ALS in the future in view of the wide range of neuroactive metabolites (e.g., D-glutamate, D-serine, sulphur amino acids etc.) that are known to be produced by bacteria in the gut and our own data showing significantly higher levels of these excitotoxins in ALS patients. This could be important in the pathogenesis of ALS. We acknowledge that established drugs such as Riluzole have modest effects on survival. However, extensive research into the mechanism of Riluzole suggests that its effects on glutamate receptors are limited and require high concentrations that are not possible in human trials (reviewed by Petrov et al, Front Aging Neurosci 2017; 9:68) . It’s become increasingly clear that the effects of riluzole are manifolds, including effects on sodium and calcium currents, as well as effects on voltage gated calcium currents and other effects on neurotransmission (Bellingham 2011, CNS Neurosci Ther 17, 4-31). Therefore, therapeutics that specifically target the glutamatergic pathway are still lacking. Our new and exciting data supports the possibility of modulating the glutamatergic pathway along the microbiota-gut-brain axis by influencing the gut microbiota composition.

Authors highlights the importance of prebiotics and postulate them as promising therapeutic molecules. In the case of n-3PUFAs, the main effect is thought to be mediated by resolvins and not by the microbiome (to my knowledge): why including them as prebiotics?

Is there any evidence for that, such as reducing mentioned toxins?

Response: In the section on possible therapies, we mention current strategies to restore eubiosis, and strategies that counteract the effects of dysbiosis. We include n-3 PUFA as these have previously been reported to alter the gut microbiota composition, which restore the gut microflora and reduce inflammation. There is no evidence yet about the effects of these therapies on levels of toxins.

Moreover, authors should consider that in doi: 10.1371/journal.pone.0061626 EPA supplementation has opposite effects of DHA. This work should also be discussed because not all n-3 PUFAS are supposed to be beneficial, at least in mouse models.

Response: We thank the reviewer for this suggestion and have added this work to section 4.2 and also included in Table 3.

There are some "-" in the middle of some words and also some characters in different font (some bacteria names).

Response: we have edited the paper to correct this. We have followed established conventions for the use of italics for bacterial names.